# Information in Streetscapes—Research on Visual Perception Information Quantity of Street Space Based on Information Entropy and Machine Learning

**Ziyi Liu [1], Xinyao Ma [1,\*], Lihui Hu [1], Shan Lu [1], Xiaomin Ye [2], Shuhang You [2], Zhe Tan [3] and Xin Li [4]**

[1]  School of Civil Engineering and Architecture, Zhejiang Sci-Tech University, Hangzhou 310018, China
[2]  Huadong Engineering Corporation Limited, Hangzhou 311122, China
[3]  College of Landscape Architecture, Nanjing Forestry University, Nanjing 210037, China
[4]  School of Information Science and Engineering, Zhejiang Sci-Tech University, Hangzhou 310018, China
[\*]  Correspondence: mxylm@zstu.edu.cn

**Abstract:** Urban street space is a critical reflection of a city's vitality and image and a critical component of urban planning. While visual perceptual information about an urban street space can reflect the composition of place elements and spatial relationships, it lacks a unified and comprehensive quantification system. It is frequently presented in the form of element proportions without accounting for realistic factors, such as occlusion, light and shadow, and materials, making it difficult for the data to accurately describe the complex information found in real scenes. The conclusions of related studies are insufficiently focused to serve as a guide for designing solutions, remaining merely theoretical paradigms. As such, this study employed semantic segmentation and information entropy models to generate four visual perceptual information quantity (VPIQ) measures of street space: (1) form; (2) line; (3) texture; and (4) color. Then, at the macro level, the streetscape coefficient of variation (SCV) and K-means cluster entropy (HCK) were proposed to quantify the street's spatial variation characteristics based on VPIQ. Additionally, we used geographically weighted regression (GWR) to investigate the relationship between VPIQ and street elements at the meso level as well as its practical application. This method can accurately and objectively describe and detect the current state of street spaces, assisting urban planners and decision-makers in making decisions about planning policies, urban regeneration schemes, and how to manage the street environment.

**Keywords:** street space; information entropy; machine learning; visual perception; spatial change; spatial heterogeneity

## 1. Introduction

Streets are an integral part of cities and citizens often perceive the urban landscape through them. Several disciplines conduct intensive research on streets because of their multifaceted characteristics [1], especially for studies on urban greenery [2,3], walkability [4,5], urban morphology [6,7], and urban perception [8,9], where they demonstrate ideal research value and potential. Urban street perception has usually been studied using manual surveys [10–14]. However, it is difficult to rely on data obtained from manual surveys for urban modeling due to the limitations of human cognition and economic conditions. Under the call of smart urbanism, using modern information and communication technology (ICT) to analyze urban data to assist in decision-making and planning will have significant competitiveness [15,16], which is one of the reasons for the surge in the number of studies in the field of smart cities in recent years [17]. The rise of artificial intelligence (AI) tools such as machine learning (ML) and deep learning (DL) provides a new direction for the realization of smart cities [18,19]. Open platform interfaces provided by companies, such as Google [20], Tencent [21], and Baidu [22], also facilitate the acquisition of big data in cities. Therefore, using street view images to study urban street space perception and

using artificial intelligence algorithms for data modeling has become one of the paradigms from the perspective of smart cities. This study is also guided by this paradigm, processing Baidu Street View (BSV) through machine learning methods and quantifying urban street perception with information entropy. Thus, exploring the visual perception of street space is useful for understanding the relationship between street elements and urban appearance, describing the built environment, or evaluating space quality, providing scientific guidance for urban planning and construction, and environmental planning.

As a widely recognized approach to urban development research, machine learning bridges the gap between artificial intelligence and urban governance [23]. Scholars can use machine learning algorithms to investigate the deeper laws underlying big data in order to explain various complex phenomena in today's urban development process [24]. With the help of such intelligent applications, urban management and policy-making are gradually being transformed into an interdisciplinary and diverse intelligence system [25]. Deeplabv3+ has been increasingly applied in urban science [26–28]. Semantic segmentation techniques for deep learning can batch process images and extract data, such as the proportion of greenery, vehicles, pedestrians, and buildings [29–31], and thus make the research results more interpretable [32–35]. They make it possible to describe the urban built environment through visual perception information [21,35–37], and argue that the ambiguity of visual information in the urban environment requires additional attention in quantitative urban appearance research [35]. Based on this, Verma enriched the perception system by exploring the development of audiovisual perception models [38], and some studies used visual entropy or partial street elements to measure the complexity of street interfaces [9,39]. Researchers have carried out large-scale perception studies, proposing guiding theories for the creation of street spaces [8,40], introducing neural network algorithms to classify sample images, or refining evaluation systems to obtain more accurate perceptual models [41,42], giving ways to improve the quality of urban street spaces based on human health and well-being at the functional level [22,31,42], and establishing links between street elements and urban microclimates [6,43]. Alternatively, at the appearance level, some visual perceptual characteristics can be used to propose methods for assessing streets' spatial quality [30,44,45] while considering the influence of spatial heterogeneity at the local scale [46,47]. Current research is increasingly focused on developing methods or models for measuring the environment via machine learning to aid urban environmental planning and management [48–50].

Previous research established the feasibility of using streetscape images to perceive urban street space and established a solid foundation for future research, which can be expanded in two directions based on shared methods and findings. (1) To account for the "information quantity" involved in visual perception, deconstructed vision methodology was incorporated, which is based on determining the proportions of each street element while considering occlusion between street elements, light and shade, color, material, and shape, allowing for a more comprehensive representation of complex factors in realistic scenes. In addition to improving the traditional algorithm used for calculating visual entropy to quantify the amount of information in street spaces in previous research, this updated algorithm reflects the advantage of delving deeper into fundamental data, allowing the enhanced visual perception index to be better suited to human perception, as well as efficiently describing and quantifying street space more precisely; (2) extending the theoretical paradigm of guiding design established in previous research provides guidance suggestions for the fine-grained creation of space, which not only provides a macro-level description of the visual perception of urban streets, but also directly contributes to the street renewal design process. Therefore, the quantification of visual perception information remains critical for future research.

To accomplish these two objectives of extended research, we propose a model that can quantify the visual perceptual information quantity (VPIQ) of street spaces as well as their spatial variation. First, we calculated the VPIQ of street images using information entropy and semantic image segmentation. We measured street space variation based on the VPIQ's

coefficient of variation and clustering entropy. The geographically weighted regression (GWR) model also allowed us to obtain the characteristics of street elements that affect local road segments with fine granularity. This method can provide objective descriptions and impact characteristics of street space visualization from macro and meso perspectives. Additionally, it can serve as a reference for urban renewal policies and customizing street design solutions.

This paper introduces the study area, data acquisition, and processing processes in Section 2. Section 3 addresses three sections: (1) four measures of VPIQ in street space based on information entropy and semantic segmentation; (2) methods for measuring spatial variation in streets based on VPIQ; and (3) methods for exploring the impact of VPIQ in streets using GWR. Section 4 presents the results of the street space change measurement and GWR model. Section 5 will discuss and conclude each of the two sections of Section 4.

## 2. Materials

### 2.1. Study Areas

This study was conducted in the sub-regions of the Hubin, Qingbo, and Xiaoying streets in the Shangcheng District of Hangzhou, China. This area is adjacent to the West Lake Scenic Area and is the old city of Hangzhou's core area (Figure 1). It contains elements such as commercial streets, business buildings, old and newly constructed residential areas, railway stations, underground lines, scenic spots, and various support facilities; it also presents strong diversity and reflects, to a certain extent, the style and function of a city. This area provided an ideal basis for this study.

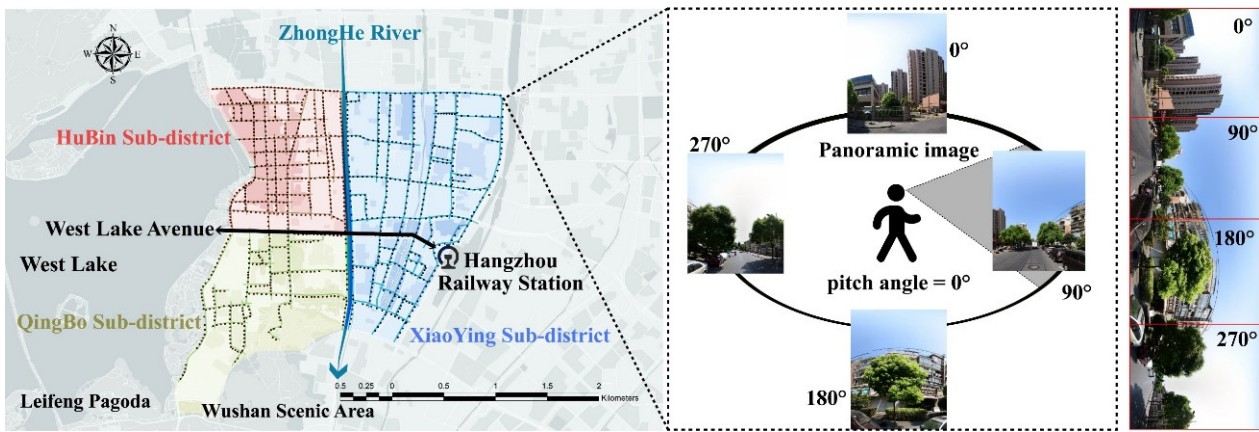

**Figure 1.** Schematic diagram of research area and panoramic image generation.

### 2.2. Data Preparation

We obtained 1179 street view images taken in September 2017 from the Baidu Maps Developer Platform (https://lbsyun.baidu.com/, access on 26 December 2021) at a distance of 40 m from streets, covering street view sampling. We built the street view image acquisition task in Python via the request library based on the uniform resource locator (URL) provided by the platform's panoramic still image application programming interface (API). Considering data privacy and security issues [17], we applied the quota of personal application keys (AK) to the platform only 100 times per day, and therefore we used multiple AK to acquire all street view images. Baidu uses confidential processing (Gaussian blurring) for human faces and license plate numbers captured in all images, which is suitable for the public publication of academic research. The horizontal and vertical field of view (FOV) range of the camera used by the Street View sampling vehicle for data collection is 80°–140°. When the pitch angle of the camera is 0°, the field of view is approximately the same as a person's normal field of view. So, we set the pitch angle to 0° when crawling the Street View images. The horizontal view angle is spliced into a panoramic view image through 0°, 90°, 180°, and 270°, which can simulate the street view of pedestrians

with different orientations (Figure 1). These images were then cleaned of duplicates and imprecise rendered images. The resolution of each image was 1024 × 512 (pixels), and the image size was 1024 × 335 (pixels) after cropping out the sampled car images and scaling equally according to different algorithms. The sampling point distribution is shown in Figure 1.

The POI data were obtained from Gaode Map (an online map company in China), and after filtering out anomalous data outside the study area, there were 9,968 records containing 15 categories.

## 3. Methods

The relationships between the calculated indicators involved in all the steps are shown in Figures 2 and 3.

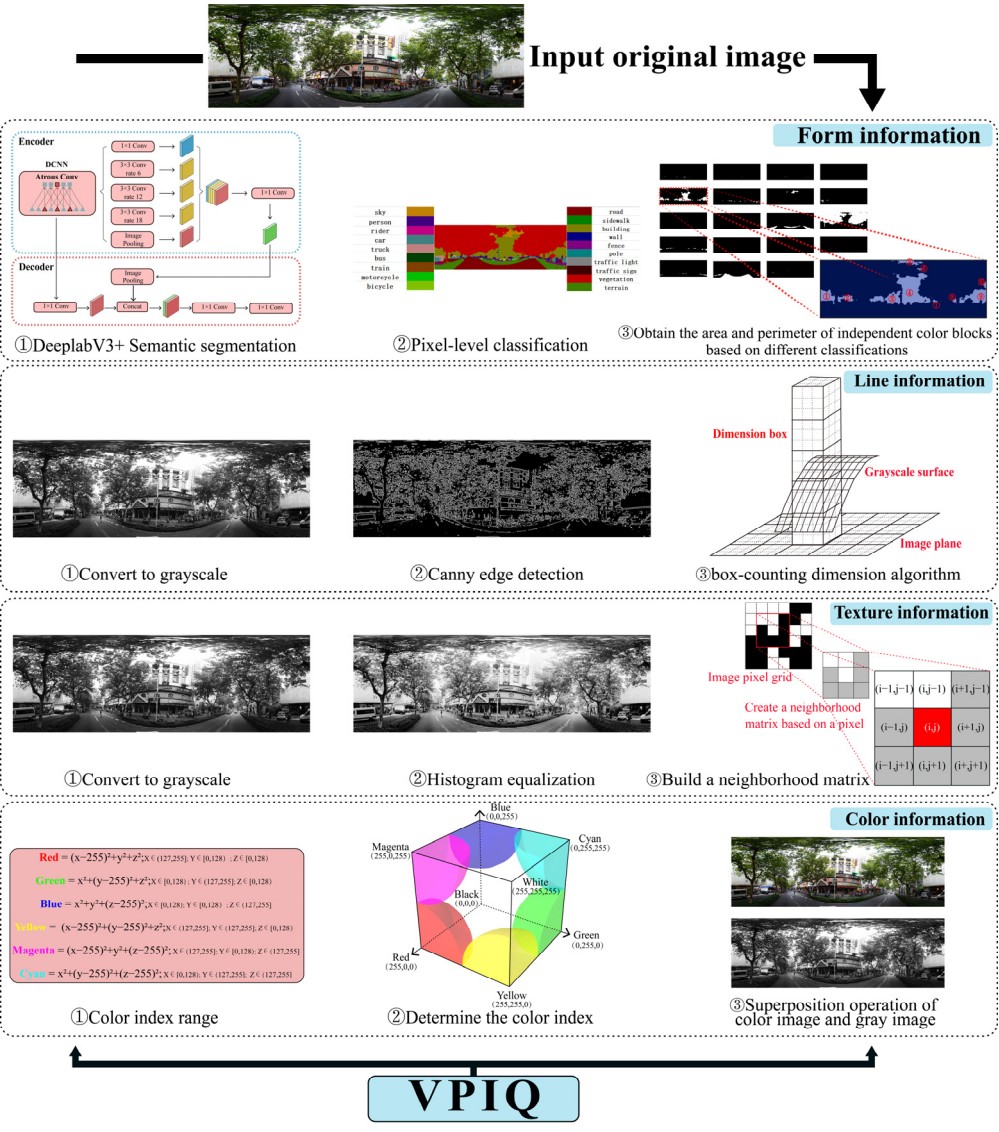

**Figure 2.** The process of visual deconstruction and the subsequent application direction.

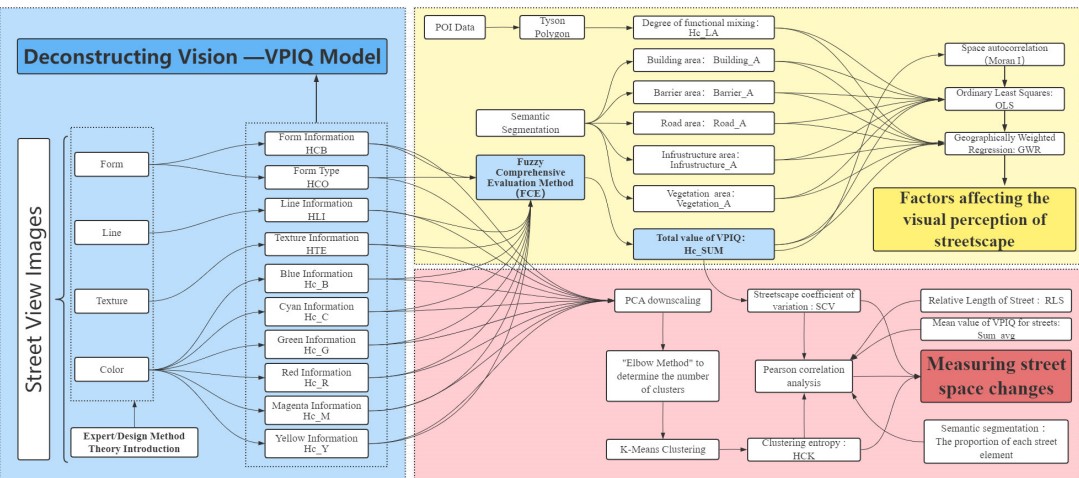

**Figure 3.** Workflow.

## 3.1. Quantification of the VPIQ Measure

The expert/design approach deconstructs vision (form, line, texture, and color) by extracting the landscape's formal features [51,52] to assess its characteristics (e.g., consistency, variety, vividness, and harmony), but this approach has been affected by excessive subjective interference [51]. However, because the use of biophysical features to describe scenery is systematically associated with visual landscape perception [53], this method of extracting abstract parameters from scenery is appropriate for studying the visual perception of landscapes.

According to the definition of "visual entropy", a landscape's visual effects are more varied and provide the human eye with more visual information when the entropy is higher. As a result, we incorporate the method of measuring landscape morphological characteristics into expert/design theory in order to decompose visual information from four perspectives: form, line, texture, and color, to quantify the visual perception information contained in the street space. Second, this study was based on an objective description of the visual information. This emphasizes the need for objective measurement over evaluation. This implies that visual information does not have the function of evaluating goodness but must be cognitively integrated with actual urban street planning and design requirements. (Figure 2).

### 3.1.1. Form

The appearance and combination of various street elements provide pedestrians with varying degrees of visual information. Rather than directly counting the percentage of street elements in the street view image, the form information defined in this paper concentrated on all the independent color blocks in the categorically labeled image obtained by semantic segmentation (Figure 2), obtained their perimeter and area using OpenCV, and calculated the information entropy of the entire image to determine how much visual information the scene's form provided. The significance of this calculation is that it accounts for the segmentation effect on the visual interface caused by mutual occlusion of street scenes (e.g., the segmentation of buildings and pavements by tree branches and utility poles), and previous studies have shown that scene occlusion has a significant impact on street interface calculation [54–56]. See Figure A1 in Appendix A for a detailed illustration.

As a result, we performed semantic segmentation of street view images using Python's Deeplabv3+ algorithm framework and the CitySpaces dataset, and then used the OpenCV tool to implement the aforementioned calculation method, and the amount of form information was formulated as:

$$HCB = -\sum_{i=1}^{n} \frac{C_i P_i}{S_i} \log P_i \qquad (1)$$

where $n$ is the total number of color blocks in the image and $P_i$ is the proportion of the $i$-th block in the image. The *HCB* is the form information quantity under consideration of the visual segmentation factor.

The richness of the types of street scenes was then added to the calculation. The semantic segmentation algorithm counts the proportion of different colors (i.e., types of street elements), which is used to calculate the complexity of the variety by formulating as follows:

$$HCO = -\sum_{i=1}^{n} P_i \log P_i \tag{2}$$

where $n$ is the total number of colors (street element classification) in the picture and $P_i$ is the proportion of the $i$-th color in the picture.

### 3.1.2. Line

Object edges, light and dark boundaries, etc. are all manifestations of line information, similar to the way a scene is recorded with pen drawing. First, the Canny edge detection algorithm was used to separate the outline and background of the scene, keeping the lines and removing other details. The Canny algorithm is a widely recognized edge detection algorithm in current research [57,58]. Then, the fractal dimension of the processed image was calculated. Among the methods for calculating fractal dimension, the box-counting dimension method is highly recognized in the planning-related fields [58]. The image was first covered with a grid matrix, where the grid side length is $\varepsilon$ and the grid number is $N(\varepsilon)$, when the grid is shrunk enough to record all $\varepsilon_n$ and $N(\varepsilon_n)$ changes, based on $\log(1/\varepsilon_n)$ and $\log(N(\varepsilon_n))$, a scatter plot was drawn and the slope HLI of the fitted line was recorded (Figure 2), which is the box-counting dimension of the graph, and its expression is:

$$HLI = \lim_{\varepsilon \to \infty} \frac{\log(N(\varepsilon_n))}{\log(1/\varepsilon_n)} \tag{3}$$

### 3.1.3. Texture

Texture information measures the variation of light and darkness on an object's surface due to unevenness, and this variation is represented as the level of grayscale values in a grayscale image, which can be viewed as a one-dimensional matrix recording different grayscale values. Two-dimensional entropy is commonly used in the field of communication engineering to calculate texture information rather than one-dimensional entropy because it can record the spatial distribution properties of each grayscale pixel in an image [59–61]. We traversed each pixel and the eight pixels surrounding it using a $3 \times 3$ grid and denoted the grayscale value of the pixel by $i$ and the mean of the grayscale values of the eight surrounding pixels by $j$, which was noted as a binary group $(i, j)$. Python was used to perform histogram equalization on the sample image to aid in the extraction of the object's texture. The probability of occurrence of $(i, j)$ in the image was then calculated and incorporated into the information entropy formula as follows:

$$j = \frac{\sum_{k=1}^{8} j(k)}{8} \tag{4}$$

$$HTE = -\sum_{i=0}^{255} \sum_{j=0}^{255} P_{ij} \log P_{ij} \tag{5}$$

where $i$ is the gray value of the $i$-th pixel in the picture ($i \in [0, 255]$), $j$ is the neighborhood gray value ($j \in [0, 255]$), and $P_{ij}$ is the probability of the binary group $(i, j)$ appearing in the picture.

### 3.1.4. Color

Color is a primary aspect of visual perception, and elements such as vegetation, vehicles, buildings, roads, and other infrastructure influence the amount of information available through their varying colors. The amount of color information in a scene can be used to quantify color richness.

We referred to Han's method for calculating the color of street scenes [62] which involved reading the RGB values from the images in MATLAB and projecting the three values into a three-dimensional coordinate system to form an RGB color cube (Figure 2). Pixel points with RGB coordinates within the specified range were filtered, and the pixel point's maximum RGB value was extracted as the color metric. The range of values for each color using the spherical equation is shown in Figure 2. The obtained color metric was substituted into the equation to compute the information entropy, yielding the following equation for the color information measure:

$$HC = -\sum_{i=1}^{n} C_i P_i \log P_i \tag{6}$$

where $C_i$ is the color metric of pixel $i$, $P_i$ is the probability of the gray value of pixel $i$ occurring in a grayscale image, and $Hc$ is the image's color information value.

To obtain a more accurate representation of a scene's color distribution, we independently counted the colors in each corner of the color cube and calculated the color information values for the six colors Hc_B (blue), Hc_G (green), Hc_C (cyan), Hc_M (magenta), Hc_R (red), and Hc_Y (yellow). Considering that the weather and time difference of each street view image can affect the image brightness, we adopted the RGB histogram method to batch correct the image brightness (Appendix B, Figure A2).

### 3.2. Exploring Spatial Variation Using VPIQ in Street Space

Historically, planning and design have frequently been conducted on a street-by-street basis. Thus, this section examines the spatial variation in the streetscape from a macro perspective, at both the street's general and internal levels, using the VPIQ values of the street space as a statistical unit.

### 3.2.1. Visual Variation in the Overall Single Street Space

Hc_SUM was calculated by aggregating the VPIQ values for 77 streets in the study area and adding HCB, HCO, HLI, HTE, Hc_G, Hc_M, Hc_R, Hc_Y, Hc_B, and Hc_C. In order to avoid the influence of different weather conditions in each street view image, when calculating Hc_B and Hc_C, the sky part was replaced with black pixels (RGB value is 0) according to the semantic segmentation results. Then, the above operators were normalized to make them comparable. Finally, the weight of each operator was determined by the fuzzy comprehensive evaluation method, and the VPIQ total value Hc_SUM was obtained by weighting.

Fuzzy Comprehensive Evaluation (FCE) is based on the membership degree theory of fuzzy mathematics, which transforms qualitative evaluation into quantitative evaluation, that is, it uses fuzzy mathematics to make an overall evaluation of things or objects that are constrained by various factors. First, we clarified the problem and established 3 hierarchical structures, including the target layer, the criterion layer with 4 factors, the sub-criteria layer and the scheme layer with 10 factors. We constructed judgment matrix $A = (a_{ij})n \times n$, $i,j = 1, 2, \ldots, n$, where $a_{ij}$ often takes values as shown in Table 1, and then performed hierarchical single sorting, that is, finding the maximum eigenvalue of judgment matrix $A$, the approximate value $\lambda_{max}$, and its corresponding characteristic equation $AW = W\lambda_{max}$ to solve the corresponding eigenvector, and then normalized its eigenvector Wi and checked the consistency. Finally, according to the total ranking of the layers, the combination consistency test was carried out, and the ranking result calculates the relative importance weight of each factor of the scheme layer to the

target layer. According to the relevant research in the literature, the adoption frequency of 10 indicators was used to calculate the indicator weights by using Yaahp software, which supports the fuzzy comprehensive evaluation method and is highly integrated with the existing AHP functions.

**Table 1.** Judgment matrix $a_{ij}$ constant value table.

| The Importance of $B_i$ Compared to $B_j$ | Absolutely Important | Very Important | Comparatively Important | Slightly Important | Same Important | Slightly Minor | Comparatively Minor | Very Minor | Absolutely Minor |
|---|---|---|---|---|---|---|---|---|---|
| $a_{ij}$ | 9 | 7 | 5 | 3 | 1 | 1/3 | 1/5 | 1/7 | 1/9 |

The streetscape coefficient of variation (*SCV*) is then calculated for each street to measure the overall dispersion of the VPIQ for each street, which is formulated as:

$$SCV = \frac{\sigma_i}{\mu_i} \tag{7}$$

where $\sigma_i$ is the standard deviation of street *i* and $\mu_i$ is the expected value of Hc_SUM in street *i*. The degree of dispersion caused by the Hc_SUM increases with *SCV* size, making the visual differences more obvious. To determine the degree of variation in the visual perception of street space, *SCV* should be used in conjunction with other indicators because, generally speaking, pedestrians do not visit a street from beginning to end (Figure 4).

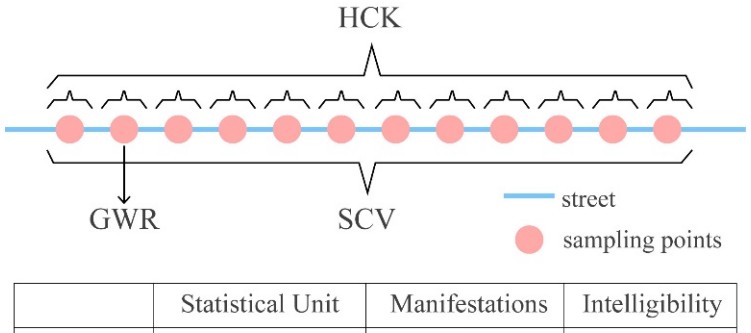

| | Statistical Unit | Manifestations | Intelligibility |
|---|---|---|---|
| SCV | Street | Street | Abstract |
| HCK | Point | Street | Intuitive |
| GWR | Point | Point | Intuitive |

**Figure 4.** Differences in statistical units and expressions of SCV, HCK, and GWR.

3.2.2. Visual Variation within a Single Street Space

We used the normalized variables HCB, HCO, HLI, HTE, Hc G, Hc M, Hc R, and Hc Y as feature values, and used the PCA algorithm in Python's sklearn package to reduce the dataset's dimensionality. The dataset was then clustered using the K-means algorithm, which can categorize street scenes with similar forms, lines, textures, and colors and assign category labels [63]. After obtaining the results, the clustering entropy *HCK* for the label counts by street is formulated as follows:

$$HCK = -\sum_{i=1}^{k} P_i \log P_i \tag{8}$$

where $k$ is the total number of clustered labels and $P_i$ refers to the proportion of the $i$-th category of labels present in the current street. We divided the streetscape into several categories based on the composition of the VPIQ sub-metrics, and then used *HCK* to measure the complexity of the streetscape categories contained in each street. This helped us to understand the degree of variation in the current street space, and the more discrete the clustered labels present in a street, the more diverse the visual perception of the street. It can measure changes within the street at a finer granularity and is therefore more accurate when combined with *SCV* to determine the nature of each street space (Figure 4).

### 3.2.3. The Relevance of Spatial Variation to Street Elements

This section examines the bivariate Pearson correlations between each street element and SCV and HCK. Considering the street as the statistical unit, we combined the mean VPIQ value for each street (Sum_avg) with the normalized number of sampling points, that is, the street relative length (RLS).

### 3.3. Exploring the Factors Influencing the Correlation between Spatial Information Values and Street Elements

Given the layout characteristics of urban streets, spatial heterogeneity is critical [47,64]. To obtain a fine-grained view of the VPIQ in street space, we investigated the spatial heterogeneity of Hc_SUM in the urban street environment from a mesoscopic perspective by employing a geographically weighted regression (GWR) model to investigate the relationship between the total value of spatial information and the proportion of each element of street space, as well as the POI information surrounding each street image sampling point.

### 3.3.1. Geographically Weighted Regression

Although the topological relationships of urban streets are heavily influenced by manual intervention, local associations persist in space and are influenced by peripheral factors [65], which is why we chose the GWR model to investigate the local relationships between VPIQ and each influencing factor. Before this, we used ordinary least squares (OLS) to investigate global regressions that did not consider geographical attributes to verify the superiority and necessity of the GWR model. The OLS and GWR equations are as follows:

$$y_i = \beta_0 + \sum_j \beta_j X_{ij} + \varepsilon_i \tag{9}$$

$$y_i = \beta_0(u_i, v_i) + \sum_j \beta_j(u_i, v_i) X_{ij} + \varepsilon_i \tag{10}$$

where $y_i$ represents the dependent variable, $\beta_j$ is the regression coefficient of the independent variable $X_{ij}$, $\varepsilon_i$ is the residual error, and $(u_i, v_i)$ represents the spatial coordinates of location $i$. GWR adds spatial information $(u_i, v_i)$ to the OLS equation, allowing each variable to possess geographical attributes. The preceding steps were performed using IBM SPSS 26 and ArcGIS 10.2.

### 3.3.2. Selection and Implementation of Independent Variables

As an independent variable, we used the proportion of the 19 categories of street elements after semantic segmentation. Given that having too many variables would impair regression model accuracy, we condensed these 19 categories into six categories (Table 2), excluding eight variables with a mobile component owing to lack of reliability in the study results [66], and this study concentrated on the stable presence of street elements. However, because the above data are purely visual, POI point data were added to quantify the businesses located near the sampling points [67].

**Table 2.** Composition and simplified results of street element variables with POI data variables.

| | Street Elements | | POI Data | |
| --- | --- | --- | --- | --- |
| | Original Categories | Simplified Categories | Original Categories | Simplified Categories |
| Reserved Variables | Building | Building_A | Transportation Facilities | Transportation |
| | Vegetation | Vegetation_A | Road | |
| | Sky | Sky_A | Car Service | |
| | Pole | | Hotels | Residence |
| | Traffic light | Infrastructure_A | Real estate communities | |
| | Traffic sign | | Recreation | |
| | Fence | | Restaurants | Entertainment |
| | Wall | Barrier_A | Tourist Attractions | |
| | Terrain | | Shopping | |
| | Road | Road_A | Life Services | Life |
| | Sidewalk | | Company | |
| | | | Finance | |
| | | | Business Building | Public |
| | | | Medical | |
| | | | Government Agencies | |
| Excluded variables | Person, Rider, Car, Truck, Bus, Train, Motorcycle, Bicycle | | | |

We collected data on all POI points within the study area and condensed the original 15 categories into five (Table 2). All sampling points within the study area were then divided into distinct areas using Tyson polygons, allowing all POI points to be projected onto a grid corresponding to each sampling point. The number of POI points within each grid was counted by category and their degree of functional mixing was computed, which can be formulated as

$$Hc\_LA = -\sum_{i=1}^{5} P_i \log P_i \tag{11}$$

where $P_i$ is the proportion of POI points in category $i$ of all POI points in the grid.

## 4. Results

### 4.1. Results of the Spatial Variation of Streets Measured Using VPIQ

This subsection examines the methods and results of explaining spatial variation in streets using VPIQ, developing the analysis in two dimensions: one from the street as a whole, measuring the intensity of changes in the total value of VPIQ, and another from within the street, examining the diversity of composition among the VPIQ sub-indicators. Finally, combining these two methods, we discuss the changes in street space.

#### 4.1.1. Overall Performance of the Single Street VPIQ

In Section 3.2.1, Hc_SUM was calculated according to the FCE method, and its weight was determined by the results presented in Table 3 after the research team had repeatedly compared and referred to previous studies. Then, the streetscape coefficient of variation (SCV) of Hc_SUM was calculated for 77 streets, and the maximum value was approximately ten times the minimum value, indicating that the variation in VPIQ is more significant between streets.

#### 4.1.2. Internal Performance of the Single Street VPIQ

This section demonstrates the performance of local visual perception on a single street using the clustering algorithm. First, we used principal component analysis (PCA), which reduced the original data matrix from eight to three dimensions while retaining 95% of the original data's information, then we used the K-means algorithm to add clustering cluster labels to the data. The optimal number of clusters was determined by "elbow method", utilizing the yellowbrick package in Python to determine, as illustrated in Figure 5, that

the dataset clusters optimally when K = 6. To visualize the distribution, we projected k-means labels onto the map (Figure 6a). The clustering algorithm revealed that the VPIQ classification of the streetscape sampling points was more pronounced, with more streetscape images in the local area classified in the same category, not just on the same street, indicating a degree of correlation in the streetscape VPIQ within the local area.

**Table 3.** Using the FCE method to determine the weight results of Hc_SUM sub-indicators.

| Target Layer | Criterion Layers and Weights | | Scheme Layers and Weights | |
|---|---|---|---|---|
| | Form information | 0.355 | HCB | 0.304 |
| | | | HCO | 0.051 |
| | Line information | 0.145 | HLI | 0.145 |
| | Texture information | 0.145 | HTE | 0.145 |
| Hc_SUM | | | Hc_B | 0.030 |
| | | | Hc_C | 0.091 |
| | | | Hc_G | 0.170 |
| | Color information | 0.355 | Hc_M | 0.011 |
| | | | Hc_R | 0.023 |
| | | | Hc_Y | 0.030 |

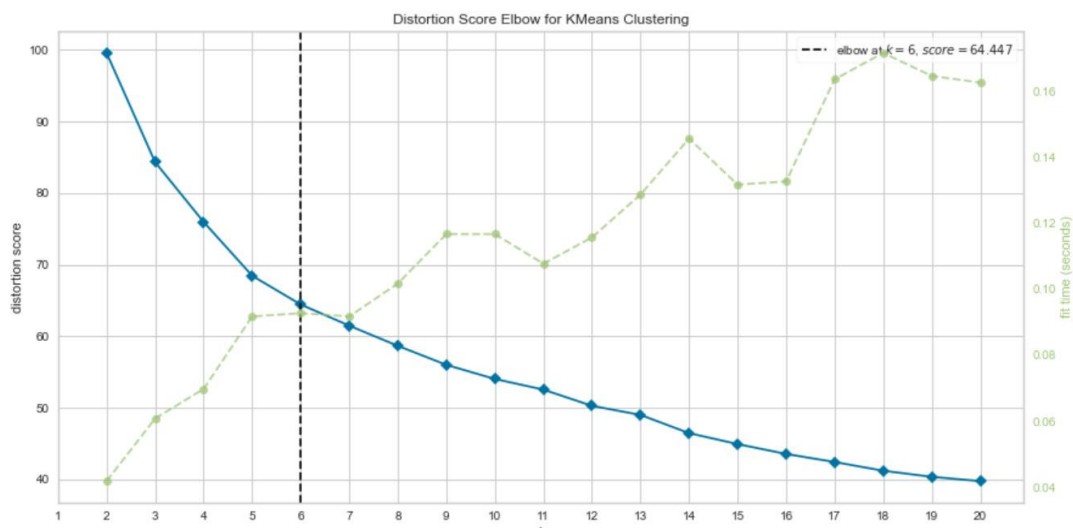

**Figure 5.** Determining the optimal number of clusters for K-Means using the elbow method.

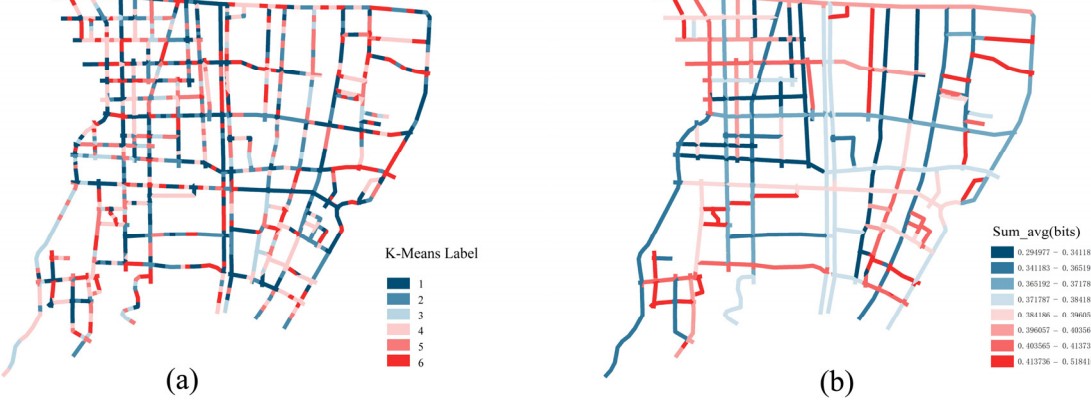

**Figure 6.** K-Means clustering label distribution (**a**); street average visual perception information (Sum_avg) distribution (**b**).

### 4.1.3. Coupling Results

The Pearson correlation analysis between the SCV and HCK in SPSS is shown in Figure 7. HCK was significantly correlated with Sum_avg, RLS, building area, walls, roads, streetlights, traffic signals, and buses. Among them, Sum_avg, building area, and walls were negatively correlated, indicating that when a greater proportion of these street elements was present, the VPIQ of the street space tended to be more consistent, resulting in a more balanced visual experience when visiting the same street. However, SCV was significantly correlated with the relative length of the street, buildings, vegetation, and sky. Only vegetation was positively correlated, indicating that streets with significant fluctuations in the total value of VPIQ (larger SCV) due to spatial variation may have a higher rate of green vision. This suggests that visual perception is more sensitive to street green space and requires strict control of the amount of vegetation.

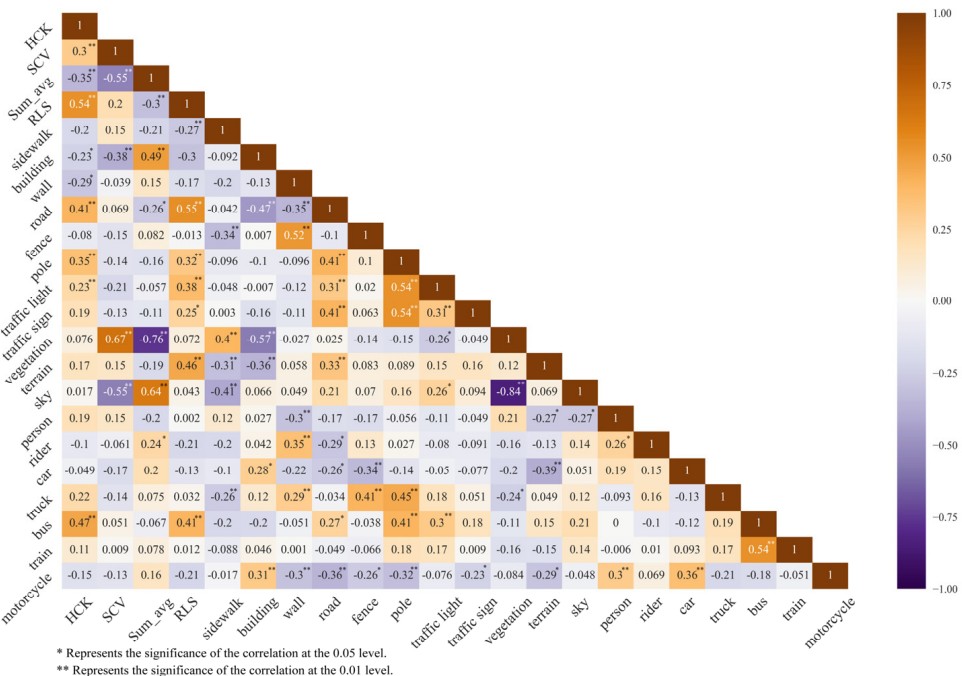

**Figure 7.** Pearson correlation matrix between variables for spatial variation studies.

In addition, we mapped Sum_avg onto this map (Figure 6b). Sum_avg indicates whether the visual perception of the current street is complex, and thus, when combined with HCK and SCV, can provide a more detailed description of the street environment's current state. On Nanshan Road, for example, where Sum_avg is low but SCV and HCK are both high, this indicates that the current street has more diverse spatial variations but a less informative visual perception, resulting in a less confusing street interface. This reflects the efforts of urban planners and managers to enhance visitor experience while retaining control over street appearance.

### 4.2. Analysis of Factors Influencing the Value of Spatial Information
#### 4.2.1. OLS Results

We began by performing a global regression on the data using OLS and recording the variance inflation factor (VIF) for each variable to determine whether any variables were multicollinear. Table 4 summarizes the identified variables and their descriptive statistics and regression results. With all variables passing the covariance diagnostic (VIF < 7.5) and residuals following a normal distribution (Appendix A), the corrected Akaike information criterion (AICc) was −3926, with $R^2$ = 0.534 and adjusted $R^2$ = 0.532, indicating that the independent variables explained 53.2% of the variation. The variable sky_A was omitted because it cannot be artificially controlled and would result in spurious correlations [46], or

because the sky itself has a poor representation of texture and color information, semantic segmentation is unable to identify clouds, and the system did not consider the sky as interpretable in the regression interpretability. The remaining six independent variables that can have a significant effect on Hc_SUM were building_A, barrier_A, road_A, infrustructure_A, vegetation_A, and Hc_LA. Their correlation coefficients were all negative correlation (Coef < 0), indicating that visual complexity decreases as the area of the first five variables increases or as the surrounding businesses of the sampling points diversify. The results above are global regression results that do not include geospatial attributes. In the following section, we compare them to the GWR results.

**Table 4.** Description of the explanatory variables and regression results for OLS and GWR.

| Independent Variable | Description | OLS | | | | GWR | | | | |
|---|---|---|---|---|---|---|---|---|---|---|
| | | Coef | t | p | VIF | Mean | std | 25% | Median | 75% |
| Building_A | The proportion of architectural elements in the picture | −0.043 | −0.675 | 0.500 | 1.627 | −0.070 | 0.275 | −0.469 | −0.082 | 0.287 |
| Vegetation_A | The proportion of vegetation elements in the picture | −1.017 | −28.881 | 0.000 *** | 1.405 | −1.077 | 0.152 | −1.368 | −1.089 | −0.827 |
| Hc_LA | The degree of mixing of business functions around the sampling point | −0.025 | −2.766 | 0.006 ** | 1.004 | −0.017 | 0.035 | −0.044 | −0.019 | 0.011 |
| Infrastructure_A | pole, traffic light, traffic sign, fence elements take up the proportion of the picture | −0.975 | −3.059 | 0.002 ** | 1.031 | −1.325 | 1.431 | −2.699 | −1.280 | 0.159 |
| Barrier_A | wall, terrain elements take up the proportion of the picture | −1.099 | −4.745 | 0.000 *** | 1.211 | −0.096 | 1.092 | −0.901 | −0.149 | 0.754 |
| Road_A | road, sidewalk elements take up the proportion of the picture | −1.186 | −6.331 | 0.000 *** | 1.262 | −1.357 | 0.782 | −2.055 | −1.430 | −0.780 |
| Constants | | 2.047 | 54.351 | 0.000 *** | | | | | | |
| | R$^2$ | 0.534 | | | | R$^2$ | 0.736 | | | |
| | Adjusted R$^2$ | 0.532 | | | | Adjusted R$^2$ | 0.710 | | | |
| | AICc | −3926 | | | | AICc | −9780 | | | |

** Represents the significance of the regression coefficient at the 0.01 level. *** Represents the significance of the regression coefficient 0.001 level.

### 4.2.2. GWR Results

The GWR model takes six variables except the sky as independent variables, and the results indicated that R$^2$ = 0.736 and adjusted R$^2$ = 0.710, respectively. This was better than the OLS model (adjusted R$^2$ = 0.532). Additionally, the AICc (GWR) = −9780, which is lower than the OLS model (AICc = −3926). Therefore, the GWR model is preferable for investigating the factors affecting Hc_SUM, and Table 4 summarizes the GWR results. Additionally, we investigated the spatial autocorrelation of Hc_SUM, which was determined by calculating the global Moran's I (Moran's I) and using it to select the Manhattan distance that was most appropriate for the city–street relationship. Moran's I = 0.373 and a z-score of 15.66 were obtained, with a *p*-value of 0.001, indicating that the VPIQ was less than 1% likely to randomly generate this clustering pattern in space, thus rejecting the null hypothesis and satisfying the spatial statistics.

## 5. Discussion

### 5.1. Interpretation and Significance of Spatial Variation Measured with VPIQ

We show all the variables involved in the fifth part in Table 5.

**Table 5.** Description of variables in the discussion.

| Name | Description |
|------|-------------|
| HCB | Form information quantity |
| HLI | Line information quantity |
| the | Texture information quantity |
| Hc | Color information quantity |
| VPIQ | HCB, HCO, HLtheHTE, Hc (Hc_G, Hc_M, Hc_R, Hc_Y, Hc_B, and Hc_C) |
| Hc_SUM | The total value of visual perception information quantity (VPIQ) based on Fuzzy Comprehensive Evaluation (FCE) |
| SCV | The fluctuation range of Hc_SUM in a street |
| HCK | K-means clustering is performed based on 10 VPIQ indicators, and the diversity of clustering categories in each street is calculated |
| Sum_avg | The average value of Hc_SUM in each street |
| RLS | Relative length of streets |

Although HCK and SCV showed a significant positive correlation, there were still many streets with high and low distributions of SCV and HCK. Thus, this section discusses four scenarios based on a combination of two measures, HCK and SCV, which correspond to the four different types of streets in the study area. Policy constraints indicated restrictions on the nature of business and land use, landscape referred to the arrangement of street elements at the street interface.

In the case of a street with high HCK and SCV (Figure 8a), this type of road reflects a lack of control over street planning, core elements, and consistent style, which should be considered in subsequent urban design. According to the correlation analysis and the actual scenario, this street type has an uncoordinated configuration of elements, frequently with an asymmetrical ratio of buildings to vegetation, resulting in a decrease in the proportion of buildings or an increase in the proportion of vegetation, as predicted by the correlation analysis. Thus, the form of vegetation has a greater visual impact than the pure greenery proportion [40]. Simultaneously, the layout of street furniture and facilities is more disorganized and unmanaged, resulting in a cluttered street interface and amplification of local indicators values of form (HCB) and lines (HLI), such as Zhonghezhong Road East, Wunsa Road, Renhe Road, and Changsheng Road. Nanshan Road is unique in that it connects the city to the scenic area, requiring its landscape to adapt to the surrounding landscape's changing nature. Although both SCV and HCK levels were elevated, Sum_avg remained low, indicating that the overall landscape was still under policy constraint. Consequently, such roads must be evaluated in segments based on site characteristics.

Streets with a low HCK and SCV (Figure 8b) are generally well-managed or subject to policy constraints and have a uniform appearance, but they can become too stagnant if they are too long (high RLS) or have a low Sum_avg. Although these roads vary in terms of scale, facilities, and adjacent businesses, they all feature attractive streetscapes, complete building façades, and a high degree of VPIQ, which corresponds to the Pearson test results. Because of the negative correlation between buildings and HCK and SCV, and the fact that the majority of these streets have a building ratio less than the mean of 20.53 percent and a harmonious relationship with the vegetation ratio, HCK and SCV were low overall. Streetlights and traffic signals are largely concealed by vegetation, and the proportion of street elements obstructing vision decreases, creating a more uniform overall feeling, as observed on Qingchun Road, Qingin Street, and Wushan Road. However, while Sum_avg is lower on Huimin Road, visual perception is more uniform but monotonous.

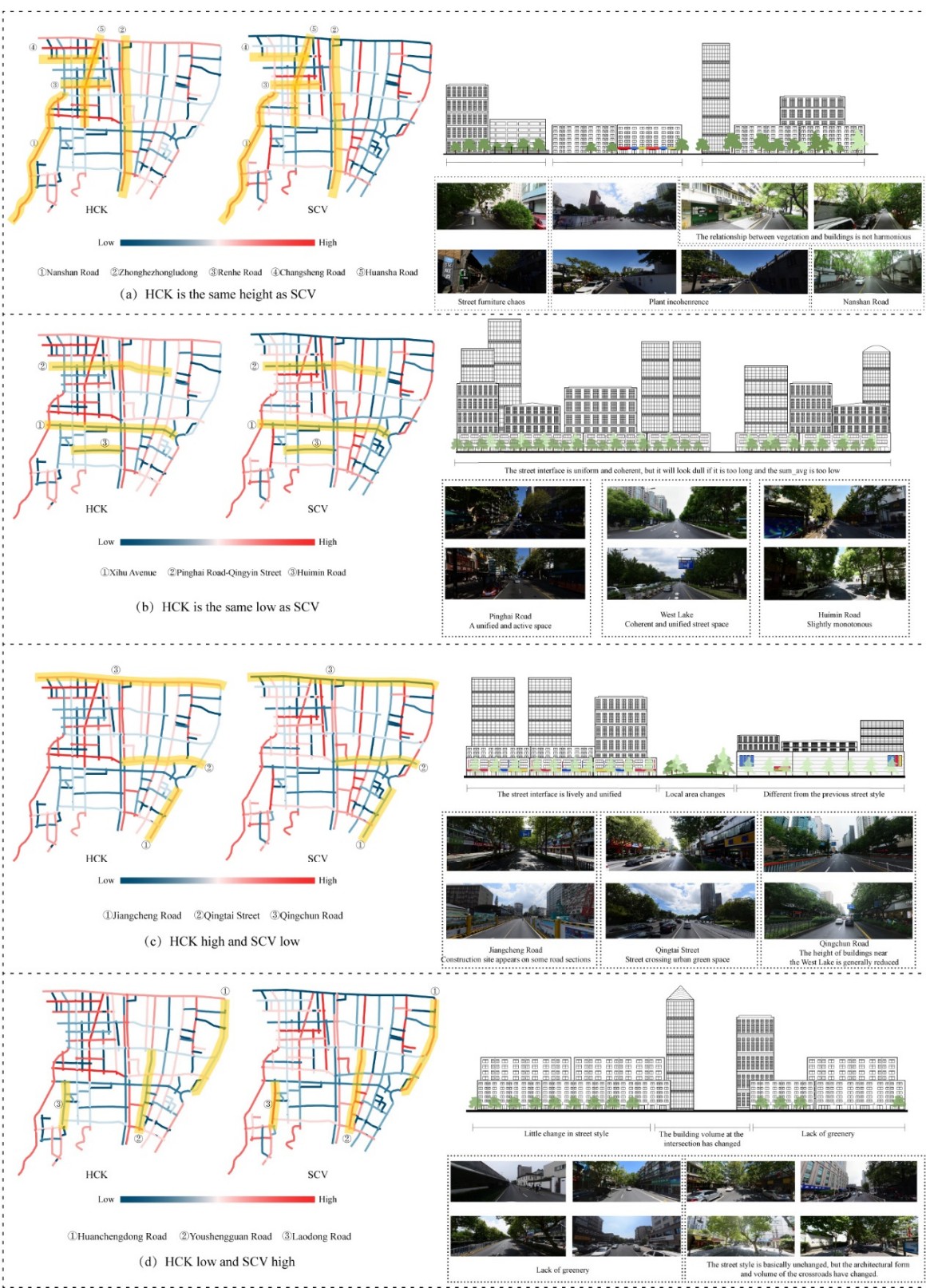

**Figure 8.** Comparative analysis of SCV and HCK, as well as elevation schematic and actual scene demonstration of typical spaces.

Streets with a high HCK and a low SCV (Figure 8c) typically indicate a change in a street section's local appearance as a result of a change in the site's nature, a change in the primary business, or an unusual street condition. The extent to which this change is beneficial must be determined in context. The impact of land use, such as a road crossing a green space, results in a change in neighborhood appearance, a change in the type of business, such as a commercial street surrounding a residential street, or an unusual condition, such as the presence of a large-scale construction site. Overall, the relationships and proportions of these streets are stable, and they all have an intact interface, which results in low SCV. However, these changes result in localized visual perception fluctuations, which increase HCK, and appropriate additions of street furniture and structures can be considered to maintain the architectural interface [66]. Jiangcheng Road, Qingtai Street, Yan'an Road, and Jianguo Middle Road are examples of such roads.

Streets with low HCK and high SCV (Figure 8d) are typically smaller-scale urban roads with relatively stable businesses and styles; as a result, they have a low HCK, but because they intersect with some major arterial roads with numerous tall buildings, tall buildings appear at road intersections. Instead of being as rich in texture and color as other buildings, these buildings are shaded by vegetation and have fewer divisions, resulting in a decrease in the 10 VPIQ indicators overall, but little change in the overall proportion, as seen in the area of Ma Shi Street–You Sheng Guan Road. However, in an older residential area, local sections may lack shops and vegetation, as evidenced by insufficient building setbacks or monotonous fences in institutional compounds, such as those on University Road and Labor Road, which may be considered for community space renewal.

We must contextualize the Pearson correlation analysis results in terms of actual space creation. For example, building proportion and wall proportion are negatively correlated with HCK, which does not imply that simply reducing the building or wall interface will result in larger changes in HCK. However, streets with higher HCK tend to have lower building and wall proportions, thus enriching their façades and increasing texture (HTE) and color information (Hc), while keeping the area of the building and wall interface unchanged, can also improve HCK. Additionally, the differing form and location of streetlights and traffic lights can affect their dividing effect on the streetscape, thereby altering the original proportion of morphology and line information. When policy permits or there is reasonable demand, the road surface can be appropriately widened, or the plant configuration of the traffic separation zone can be enhanced to alter the road's form (HCB) and line information (HLI). All of these measures have the potential to alter the VPIQ of street spaces.

In summary, streets with a high SCV are more likely to have anomalies, which usually indicate that the street interface is confusing or incomplete, whereas HCK interpretation is based on the actual situation to determine the reasonableness of the current street interface. If SCV and HCK are both high or low when combined with Sum_avg, this determines whether the current road requires improvement. The importance of using the VPIQ to measure consistency is that it allows urban planners and management decision-makers to quickly understand the current status of each street in data form and consider how to tailor constructive planning solutions based on planning policies and demand positioning.

### 5.2. Significance and Application of Factors Affecting VPIQ

Each of the explanatory variables is discussed in terms of how it affects Hc_SUM and how this information is understood and applied. This calculation is based on the GWR model output, which provides planners with more detailed measurement results. This demonstrates how different sections of the same street are differently affected by these factors. Consequently, it can serve as a reference for more refined design solutions. The relationship between the influence of each explanatory variable and Hc_SUM in the GWR is shown in Figure 9.

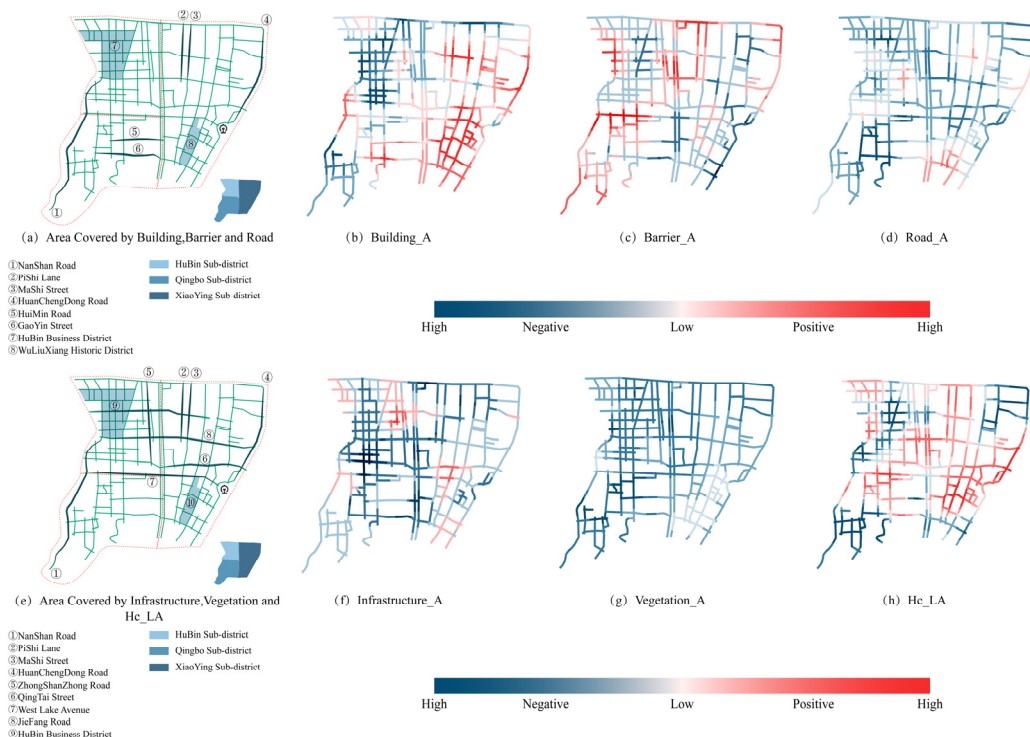

**Figure 9.** GWR model results, where subplot (**a**) labels typical streets and areas in variables Building_A, Barrier_A, and Road_A, while subplot (**e**) labels typical streets and areas in variables Infrustructure_A, Vegetation_A, Hc_LA. The darker blue color represents the stronger negative correlation, while the darker red color represents the stronger positive correlation.

In the case of Building_A (Figure 9b), a negative correlation area typically indicates that the mass or style of the building is somewhat controlled, and thus contributes less visual information. For example, large commercial complexes and office buildings with a uniform façade in the Hubin sub-region will be more minimalist than city streets with brightly colored signs [68], resulting in a lower Hc_SUM. By contrast, positively correlated areas are frequently defined by disparate building masses, richer façades, and increased obscuration and division by other street elements, allowing architectural elements to contribute more visual information, as evidenced by the abundance of shopfront signage and windows in Wuliu Xiang historic district. Therefore, the regression result of Building_A can measure the contribution of buildings in this area to the amount of visual perception information. When the street interface is too chaotic, it can be judged that the building façade needs to be simplified or enriched according to the positive and negative factors and the size of the building_A variable in the current area or combined with other variables to obtain a more comprehensive construction strategy.

According to the quartiles of the GWR regression coefficients (Figure 9c), Barrier_A (mean = −0.096) has a greater influence on Hc_SUM than Building_A (mean = −0.070), whereas the segmentation of the city spaces dataset indicates that fences and some street furniture are likely to be classified as walls or terrains, indicating that these elements are sufficiently important in the streetscape composition. The positive correlation indicates that these areas have more structures built by residents and businesses, and that these structures with street furniture have more diverse forms (high HCB) and lines (high HLI) or distinctive textures (high HTE), and thus have a significant positive effect on Hc_SUM [66], such as the northwest corner of Hubin sub-region, Qingbo sub-region, and Huancheng East Road, Pi Shi Lane, and Ma Shi Street in Xiaoying sub-region The negative correlation is most likely due to the presence of construction sites or institutional walls in the study area, which are typically bland in form (low HCB) and monotonous in texture (low HTE)

and color (low Hc), and thus contribute little to Hc_SUM. Therefore, the positive and negative correlations of the GWR model in the Barrier_A variable reveal the visual impact of the fence and street furniture in the current area, the management of street furniture and structures in the positive correlation area, and the lost space needs outside the courtyard fence in the negative correlation area receive attention.

In the case of Road_A (Figure 9d), there is a predominantly negative correlation (mean = −1.357) in the study area, which is due to vehicle shading and the road's lack of distinct texture (HTE) and color (Hc), but also because streets with large areas typically have more open space and are further away from elements such as buildings and vegetation, diluting the texture and color details. The road traffic condition may be improved in the positive correlation area, the interval distance moderates the traffic flow on the road, creating the effect of separation, and an open work fence separates the road in the middle, increasing the form (HCB) and line information (HLI). As a result, streets with high Road_A values should pay more attention to the creation of backbone elements, such as buildings and vegetation, while for areas with strong negative correlation, the visual impact of their traffic conditions should also be considered in the design scheme.

The mean value of the regression coefficient in Infrustructure_A (Figure 9f) demonstrated a strong negative correlation (−1.325). This appears to contradict the perception that streetlights and fences divide the streetscape, increasing Hc_SUM. However, in reality, most streetlights are obscured by vegetation, and fences are obscured by vehicles. These elements lack rich color (high Hc) and texture information (high the), while traffic signals only take up an insignificant proportion of the streetscape. It implies that road facilities, such as Jiefang Road and Yan'an Road, do not contribute as much to Hc_SUM as theoretically predicted. Rather, elements, such as light poles on streets in positively correlated areas, such as Huan Cheng Dong Road, Qing Tai Street, and the area surrounding Ping Hai Road-Qing Yin Street, are typically not obscured by trees. Streets in positively related areas should pay attention to the relationship between street facilities and the surrounding environment. If the current environment is too chaotic, it is necessary to reasonably choose the shape and volume of facilities to avoid adding unnecessary visual information, while streets in negatively related areas can consider enriching the appearance of street facilities by using artistic materials, shaped fences, streetlights, etc. if they have the need to enhance visual vitality.

Vegetation_A (Figure 9g) exhibits a strong negative correlation with the overall value (−1.077), indicating that it mainly plays a more covering role in the streetscape. Additionally, the sampling was mostly cloudy, resulting in a dull vegetation color (low Hc). The shadows cast by tall buildings obscured vegetation's texture (low HTE) and color (low Hc), as well as its monotony, or it was too dense to block the sun, as in the Pinghai and Nanshan roads. Some streets with less vegetation, such as Pi Shi Lane, also exhibited a negative correlation. Excessive or insufficient vegetation can affect Hc_SUM, and previous research has reached the same conclusion [39]. Light is a significant factor in the visual perception of vegetation, and designers should pay attention to whether the vegetation in the streets of the areas with strong negative correlation affects the vision too much, and promptly prune the vegetation, appropriately replant colorful foliage plants, or consider the greening of the building façade.

The regression coefficient for Hc_LA (Figure 9h) is less significant (mean = −0.017) than that for Building_A, most likely because of some overlap in the information explained by the two variables. The positive correlation is more pronounced in areas with a diverse mix of businesses and prominent building façades. The variety of buildings, windows, and signs along the street contributes a wealth of visual information. As can be seen, a diverse mix of businesses frequently results in a more visually appealing street interface [68], increasing visual information; for instance, in the commercial area of the Hubin sub-region, the entirety of Jiefang Road, the railway station, and Wuliu Xiang areas. In contrast, business density is generally sparse and homogeneous in negatively related areas, such as the southern section of Nanshan Road, the northern section of Zhonghezhong Road, and

the northeast corner of Xiaoying Street. The regression results of Hc_LA can reflect to some extent whether the business dynamics of the location match with the street appearance, so as to measure the reasonableness of the store appearance and street interface. Areas with strong positive and negative correlation will be extra sensitive to the matching of street business positioning and style atmosphere, and should be used as a reference in the pre-planning stage.

The GWR model enables urban planners and decision makers to gain a more detailed understanding of the factors and characteristics that influence the change in Hc_SUM in a local area. Because the same street elements frequently have varying effects on the Hc_SUM of different street sections, the GWR model will aid in the development of specific plans by designers to focus on street elements that require additional attention to achieve targeted results.

*5.3. Summary*

This study followed the urban planning and design process, beginning with a macro-level discussion of the interpretation and significance of street spatial variation measurements, revealing their two primary functions: detection and reference, which aid in quantifying the current situation and planning development. This was followed by a meso-level discussion on how each street element in a local area affects the VPIQ, highlighting street elements that require additional attention in design and possible problems. The VPIQ model is applicable throughout the process of urban planning and design and provides scientific guidance.

## 6. Conclusions

The purpose of this study is to propose a method for quantifying the amount of visual perceptual information in street spaces (VPIQ) and to apply it to Hangzhou, China's old city, adjacent to West Lake. We demonstrated the feasibility of using the VPIQ to quantify spatial variation in street spaces and to interpret the visual perception influenced by street elements through empirical analysis. This will increase the efficiency and accuracy of the environmental control and spatial creation processes for urban streets.

Given that different positions evaluate the VPIQ measure differently, both in terms of macro-level streetscape control and meso-level perception of neighborhood street scenes, the approach presented here requires flexibility in its application by urban decision makers and planners. After the development of a plan for road space creation based on the results of spatial variation measurements, the GWR model continues to guide the selection of street elements to be considered when developing a design plan.

The following research limitations exist in this study for objective reasons: (1) the weather conditions caused inconsistent sky elements in streetscape images, obstructing the measurement of blue and cyan color information; (2) API limits the efficiency of streetscape image acquisition, increasing the difficulty of conducting large-scale studies.

As a result, future work will focus on the following topics: (1) utilizing image color correction techniques to ensure the integrity of color measurements; (2) expanding data access to include streetscape images from new towns and suburbs in future studies, thereby making the model sufficiently universal; (3) developing new algorithms to replace traditional algorithms for calculating texture and color information to avoid loss of accuracy and maximize research efficiency; (4) including time series in the calculation of spatial variation to simulate pedestrian perception.

**Author Contributions:** Conceptualization, Ziyi Liu and Xinyao Ma; data curation, Ziyi Liu; methodology, Ziyi Liu, Lihui Hu, Xin Li and Zhe Tan; supervision, Xinyao Ma, Lihui Hu, Shan Lu and Xiaomin Ye; validation, Shan Lu and Shuhang You; writing—original draft, Ziyi Liu; Writing—review and editing, Ziyi Liu, Xinyao Ma, Lihui Hu, Xiaomin Ye, and Shuhang You All authors have read and agreed to the published version of the manuscript.

**Funding:** This research received no external funding.

**Institutional Review Board Statement:** Not applicable.

**Informed Consent Statement:** Not applicable.

**Data Availability Statement:** Not applicable.

**Conflicts of Interest:** The authors declare no conflict of interest.

## Appendix A. Illustration of Quantization Methods for Occlusion and Segmentation

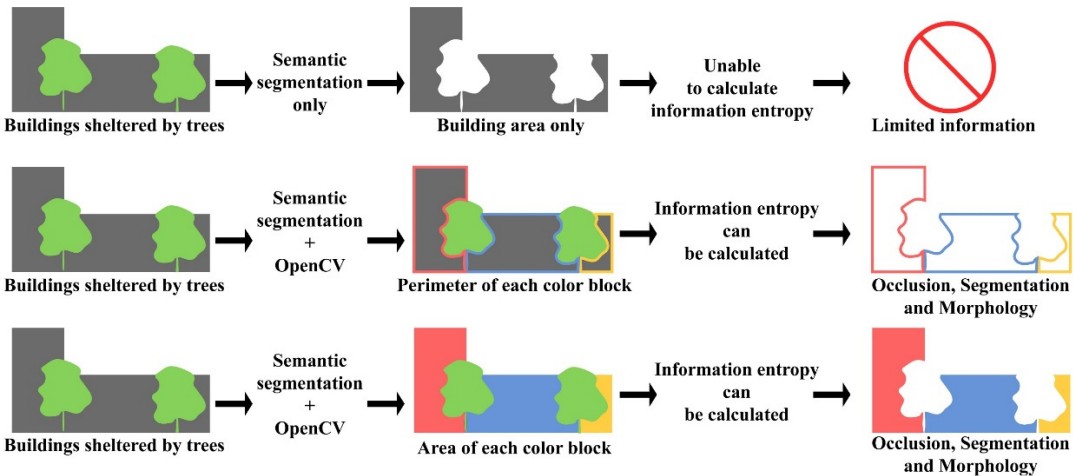

**Figure A1.** Taking the occlusion and segmentation of buildings by vegetation as an example, compare semantic segmentation statistics and HCB calculation methods.

## Appendix B. Image Brightness Correction Display

The principle is to judge the color distribution of an image by obtaining histograms of RGB channels for each street view. Generally speaking, the higher the R, G, and B values (the upper limit is 255), the more likely it is that the image will have higher brightness. The two quantile values are then set as the upper and lower limits of the RGB values in the main part of the image. Then, the values outside the quantile value interval were removed, and then the quantile value interval was stretched to $255 \times 0.1$ to $255 \times 0.9$ (to avoid pixel value overflow), thereby correcting the image brightness. For images of sufficient brightness, a judgment condition of not correcting the brightness of the image was added. In order to easily identify the sky part, we used the original image in the "before" part.

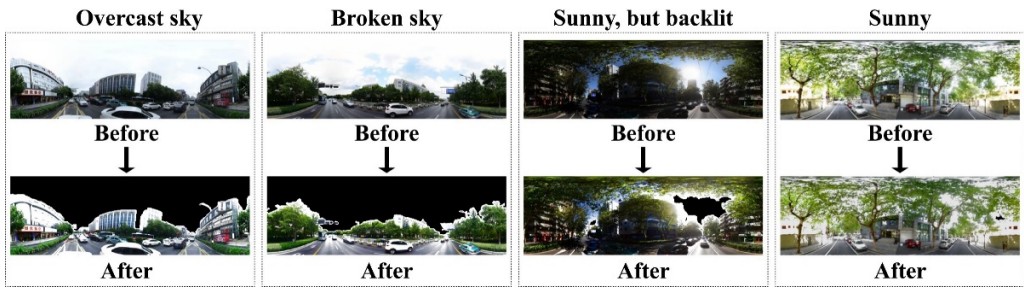

**Figure A2.** Brightness correction display for scenes with different lighting conditions.

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
