# Peer review of "Information in Streetscapes—Research on Visual Perception Information Quantity of Street Space Based on Information Entropy and Machine Learning"

_ijgi, doi:10.3390/ijgi11120628_

Round 1

Reviewer 1 Report

The manuscript is very meaningful and can be published after the details are modified.

Author Response

Dear reviewer,

We are very honored to have your endorsement of this research. We optimized the full-text content sentence by sentence. Thanks again for your endorsement!

Best regards!

Reviewer 2 Report

This is a good paper which is almost ready to be published, but some minor revisions are necessary.

First the paper needs to be better contextualised within the urban planning/studies literature. What the authors are talking about is essentially smart urbanism, so they should add at least a short paragraph on smart cities and how their findings contribute to make cities smarter.

Second the paper is about machine learning applied for urban development, but relevant contextual academic literature is missing. There is an important body of literature about the growing role of AI and machine learning in urban development. I recommend this articles:

Ullah, Z., Al-Turjman, F., Mostarda, L., & Gagliardi, R. (2020). Applications of artificial intelligence and machine learning in smart cities. Computer Communications, 154, 313-323.

Cugurullo, F. (2020). Urban artificial intelligence: From automation to autonomy in the smart city. Frontiers in Sustainable Cities, 2, 38.

Allam, Z., & Dhunny, Z. A. (2019). On big data, artificial intelligence and smart cities. Cities, 89, 80-91.

Third the authors should say more about where exactly the images came from, how Baidu collected them and if there is any issue of privacy and personal data that planners should be aware of.

Reviewer 3 Report

Summary: Authors detail the visual information in the streets to include various high and low-level feature sets, including form, line, texture, and color. The study seems exhaustive and covers the literature well. The literature and discussion are sufficient; however, a few details are challenging to understand in one sitting.

Pointwise Comments

The research points out that occlusion is a significant limitation that previous studies have also incurred; however, the authors fail to address the problem, rightly so, due to inherent problems as to how street view image datasets are collected.

Again, the use of colors depends on the time of the day and the shadows cast upon the surfaces of the materials, which would provide less accurate color representation. Similar to the above, this also cannot have an alternative as street view images from most providers do not account for time.

Is human FOV been considered in obtaining the street view? While it's common in urban sciences studies to consider custom sizes depending upon the imagery provider, there still should be scope for discussion in the text.

The images and the details regarding figures, images, statistical analysis, and amount of variables taken into consideration are immense. However, it also creates difficulty in understanding important minute details.
